# Genome-Wide Analysis of Ribosomal Protein GhRPS6 and Its Role in Cotton Verticillium Wilt Resistance

**DOI:** 10.3390/ijms22041795

**Published:** 2021-02-11

**Authors:** Dandan Zhu, Xiangyue Zhang, Jinglong Zhou, Yajie Wu, Xiaojian Zhang, Zili Feng, Feng Wei, Lihong Zhao, Yalin Zhang, Yongqiang Shi, Hongjie Feng, Heqin Zhu

**Affiliations:** 1Zhengzhou Research Base, State Key Laboratory of Cotton Biology, School of Life Science, Zhengzhou University, Zhengzhou 450001, China; zhudandan0815@163.com; 2State Key Laboratory of Cotton Biology, Institute of Cotton Research, Chinese Academy of Agricultural Sciences, Anyang 455000, China; Zhangxy9641@163.com (X.Z.); zhoujl510@163.com (J.Z.); fengzili@caas.cn (Z.F.); weifeng0108@163.com (F.W.); zhaolihongqq@163.com (L.Z.); yalinzhang2012@163.com (Y.Z.); 13619832007@163.com (Y.S.); 3School of Agricultural Sciences, Zhengzhou University, Zhengzhou 450001, China; wuyajie1213@163.com (Y.W.); xiaojianzhang993@163.com (X.Z.)

**Keywords:** cotton, Verticillium wilt, ribosomal protein, phosphorylation

## Abstract

Verticillium wilt is threatening the world’s cotton production. The pathogenic fungus *Verticillium dahliae* can survive in the soil in the form of microsclerotia for a long time, colonize through the root of cotton, and invade into vascular bundles, causing yellowing and wilting of cotton leaves, and in serious cases, leading to plant death. Breeding resistant varieties is the most economical and effective method to control Verticillium wilt. In previous studies, proteomic analysis was carried out on different cotton varieties inoculated with *V. dahliae* strain Vd080. It was found that GhRPS6 was phosphorylated after inoculation, and the phosphorylation level in resistant cultivars was 1.5 times than that in susceptible cultivars. In this study, knockdown of *GhRPS6* expression results in the reduction of SA and JA content, and suppresses a series of defensive response, enhancing cotton plants susceptibility to *V. dahliae*. Overexpression in Arabidopsis *thaliana* transgenic plants was found to be more resistant to *V. dahliae*. Further, serines at 237 and 240 were mutated to phenylalanine, respectively and jointly. The transgenic *Arabidopsis* plants demonstrated that seri-237 compromised the plant resistance to *V. dahliae*. Subcellular localization in *Nicotiana benthamiana* showed that GhRPS6 was localized in the nucleus. Additionally, the pathogen inoculation and phosphorylation site mutation did not change its localization. These results indicate that GhRPS6 is a potential molecular target for improving resistance to Verticillium wilt in cotton. This lays a foundation for breeding disease-resistant varieties.

## 1. Introduction

Cotton is one of the most important cash crops in the world and can be used as an important raw material for the textile, chemical, pharmaceutical, and national defense industries [1]. The soil-borne fungus *Verticillium dahliae* is one of the most destructive fungi in the world’s cotton growing areas [2]. The fungus infects over 200 dicotyledon plant species and live in soil as a dormant structure called microsclerotia for many years. It generally causes plant dysplasia, leaf wilt, and yellowing and browning of vascular bundles, eventually leading to early death in some plants [3,4]. Verticillium wilt causes more than 30% of China’s annual cotton output to be lost, and attempts at biological control and crop rotation have had little success. At present, breeding and screening disease-resistant varieties has become the most economical and effective mean of control [5,6,7,8,9,10]. Thus, it is essential to identify Verticillium wilt resistance genes in cotton germplasm and incorporate them into elite cotton cultivars [11].

Disease resistance is one of the most complicated yet important plant traits [12]. Although the resistance of different cotton varieties to different *V. dahliae* strains was different, the resistance of three cotton cultivars in the genus of cotton showed that sea-island cotton had stronger resistance to *V. dahliae*, followed by upland cotton, and Asiatic cotton had poorer resistance [13]. Many cellular processes have been reported to be involved in cotton resistance to *V. dahliae*, including cell wall strengthening, production of secondary metabolites, and hormone-mediated signaling pathway induction [14,15]. It is reported that Salicylic acid-related cotton (*Gossypium arboreum*) ribosomal protein GaRPL18 contributes to resistance to *V. dahliae* [16]. Meanwhile, an ethylene response-related factor, GbERF1-like, from *Gossypium barbadense* was proven to improve resistance to *V. dahliae* by activating lignin synthesis [17]. It was demonstrated that HyPRP1 performs a role in negatively regulating cotton resistance to *V. dahliae* via the thickening of cell walls and ROS accumulation [14].

The ribosome catalyzes protein translation in all organisms. It is a ribonucleoprotein particle of 70 svedberg units (70S) in prokaryotes composed of two subunits (30S and 50S in prokaryotes) [18]. The higher eukaryotic ribosomes are composed of two subunits designated as 40S (small) and 60S (large) subunits. The large subunit is composed of 25–28S, 5.8S, and 5S ribosomal RNA (rRNA) together with about 47 ribosomal proteins (r-proteins), whereas the small subunit is composed of 18S rRNA and about 33 r-proteins. The ribosomal protein S6 (RPS6), located in the small head region of the cytoplasmic 40S ribosome subunit [19], is the first and only protein shown to undergo inducible phosphorylation [20].

Post-translational modifications play a key role in many cellular processes and regulate protein activity, targeting, or interactions with other proteins [21]. Reversible phosphorylation of proteins plays an important role in the regulation of almost all life activities [22], such as gene transcription, expression, cell proliferation, differentiation, apoptosis, signal transduction, immune regulation, tumorigenesis, etc. [23]. In *Arabidopsis thaliana*, phosphorylation of ribosomal protein RPS6 integrates light signals and circadian clock signals [24]. Early changes in RPS6 phosphorylation and BH3 profiling predict response to chemotherapy in AML cells [25]. Protein phosphorylation and dephosphorylation are involved in a variety of signaling pathways. Lipid peroxidation product 4-Hydroxy-trans-2-nonenal (HNE) causes protein synthesis in cardiac myocytes via activated mTORC1-P70S6K-RPS6 signaling [26]. Increased lipogenesis, induced by AKT-mTORC1-RPS6 signaling, promotes development of human hepatocellular carcinoma [27].

In order to study the molecular mechanism of resistance to Verticillium wilt of cotton, Lquid chromatography-tandem mass spectrometry analysis (LC-MS/MS) of protein modification in root of resistant and susceptible cultivars was carried out [28]. Phosphorylation omics identification showed that the phosphorylation level of RPS6 in cotton resistant cultivars and susceptible cultivars was significantly increased after being infected by *V. dahliae*, and the phosphorylation level of RPS6 in resistant cultivars was 1.5 times higher than that of susceptible cultivars. When *GhRPS6* was silenced in cotton disease-resistant varieties, the resistance of cotton was decreased compared with that of the control group. After the phosphorylation site mutation of GhRPS6, the overexpression vector was constructed and the GFP label was attached, and the overexpression in *A. thaliana* and the cellular localization in *N. benthamiana* were carried out in order to find the key phosphorylation sites. Overexpression of *GhRPS6* in *A. thaliana* was found to increase disease resistance in transgenic plants. Our results provide an important basis for suggesting that *GhRPS6* is a positive regulator of Verticillium wilt resistance in cotton, as a potential molecular target for improving resistance to Verticillium wilt in cotton.

## 2. Results

### 2.1. GhRPS6 Gene Cloning and Bioinformatics Analysis

In the early proteomic analysis, it was found that a certain peptide segment was phosphorylated, and the peptide segment was identified as GhRPS6 (Gh_D13G0519.1) after BLAST by using amino acid sequence. It was previously reported that the ancestors of *G. arboreum* and *G. herbaceum* provided the A subgenome for the modern cultivated allotetraploid cotton [29]. While a draft genome of *G. raimondii* demonstrated whose progenitor is the putative contributor of the D subgenome to the economically important fiber-producing cotton species *G. hirsutum* and *G. barbadense* [30]. In order to study the evolutionary relationship of RPS6 in *A. thaliana*, *G. hirsutum*, *G. raimondii*, *G. arboreum*, and *G. barbadense*, the RPS6 amino acid sequence was used to construct the evolutionary tree. The results showed that GhRPS6 (Gh_D13G0519.1) of *G. hirsutum* was in the same branch as GOBAR DD31969 of *G. barbadense*, and the protein sequence was closest to that of *A. thaliana* AT4G31700 (Figure 1a). Upland cotton (*G. hirsutum*) is the most widely cultivated species globally [31]. To better understand the evolutionary relationship between different members of the *GhRPS6* gene family, we constructed separate rootless phylogenetic trees using *GhRPS6* DNA sequences and performed comparative analysis of intron–exon structures (Figure 1b,d). The results showed that the *GhRPS6* gene sequence length was significantly different, with the maximum length exceeding 10kb and the minimum length less than 2kb. The *GhRPS6* gene family is rich in intron distribution, and the intron distribution of genes with similar sequences is more similar. The conserved motifs in GhRPS6 protein were identified using MEME software. A total of 13 conserved motifs were identified in GhRPS6, named motifs 1 through 13, and most members had 1, 2, 3, 7, 11, 12, 13, and 14 motifs, indicating that they were highly conserved in GhRPS6 (Figure 1c).

To further study the *GhRPS6* gene family relationships, chromosome localization and collinearity analysis were performed on a total of 15 *GhRPS6* genes. The results showed that *GhRPS6* was distributed in 12 chromosomes of subgenome D and A, and gene repeatability found that Gh_D13G0519.1 had three duplicates, which were distributed in chromosome A04, A13, and D04 (Appendix A). In order to explore the expression pattern of *GhRPS6*, the expression spectrum of *GhRPS6* family was analyzed using transcriptional database. The results showed that Gh_D13G0519.1 and the three duplicated genes expressed the same amount in different tissues of uplands (Appendix A). The results of the expression profile after inoculation showed that the Gh_D13G0519.1 gene expression was down-regulated by *V. dahliae* (Appendix A).

*GhRPS6* was amplified from cDNA in the root of cotton *cv.*zhongzhimian2, and compared with the RPS6 sequence in the cotton database (Gh_D13G0519.1), the sequence similarity was found to be 99.34%. It was found that alanine (Ala) at the 80th position of GhRPS6 was missing, and amino acids at the 171st and 172nd positions were identical to the sequence published in the database due to the degeneracy of codon (Appendix A). It was found that GhRPS6 contains the typical S6e domain by predicting the conserved domain of the protein (Appendix A). The online database was used to predict the secondary and tertiary structures of GhRPS6 (Appendix A). Hydrophobicity analysis showed that the protein was hydrophilic (Appendix A) and did not contain transmembrane domain (Appendix A). Phosphorylation site prediction and mass spectrometry analysis showed that ser237/240 was a possible phosphorylation site (Appendix A).

### 2.2. GhRPS6 Silencingreduced the Resistance of Cotton to V. dahliae

In order to explore the role of *GhRPS6* in cotton resistance to *V. dahliae*, *GhRPS6* silenced plants were constructed by Virus-induced gene silencing (VIGS), which uses the modified viral genome to knock out the target gene in a moment, and has been widely used to decipher the function of plant genes [32,33,34]. To further verify the feasibility of this method, *GhPDS* was selected as the visible gene marker, *Agrobacteria*-GV3101 containing PYL*-GhPDS* plasmid was injected into cotton cotyledon at 10 days of age, and the albino phenotype appeared 2 weeks later (Figure 2a). *TRV:GhRPS6* plants were constructed by the same method. In addition, gene expression in the silenced plants was detected by qRT-PCR. *GhPRPS6* expression was significantly decreased in the silenced plants compared with the control (Figure 2b). The pathogen Vd080 was inoculated by root dipping method, and the phenotype was observed. Compared with the control plants, the morbidity of silencing plants was higher. Additionally, the leaf wilting and yellowing in *TRV:GhRPS6* plants were more serious, and vascular bundle browning was significantly changed (Figure 2c,d). Phloroglucinol staining showed that the xylem of the control and silencing plants was thickened after inoculation with *V. dahliae*, but the xylem of the silencing plants was thinner than the control plants (Figure 2f). The fungal recovery assay from stem sections of inoculated cotton plants also confirmed that the fungus colonization rate was faster (Figure 2g). Compared with the control group, a decreased density of callose depositions (number per cm^2^) and more dead cells was visualized in the true leaves of silencing plants (Figure 2e,h,i). These results showed that *GhRPS6* silencing reduced the resistance of cotton to *V. dahliae*, indicating that *GhRPS6* played a positive regulatory role in plant resistance to pathogenic fungus infection.

### 2.3. Expression Levels of Disease-Resistant Genes

In order to explore the effect of *GhRPS6* silencing on plant disease resistance at the transcriptional level, expression of six plant disease resistance-related genes were measured using quantitative reverse transcription PCR (RT-qPCR). The results showed that the expression levels of these PR genes in silenced plants were significantly lower than that in control plants (Figure 3). The lowest level of *GhCHI* expression was at 3 and 6 h post inoculation (hpi) in the silenced plants, which was 4.5 and 2.5 times lower than that in the control plants. PAL is the first protein and an important rate-limiting enzyme in the secondary metabolism of catalyzing phenylpropane, which is closely related to the synthesis of plant antitoxins and phenolic compounds [35,36]. The expression of *GhPAL* was quickly lowered by 4 times at 1 hpi in the silenced plants. Although the expression of *GhPAL* was increased, it was also lower than control plants. Jasmonate ZIM (Zinc-finger Inflorescence meristem) domain (JAZ) is a transcriptional inhibitor of the Jasmonic-acid signaling pathway in plants. The expression of *GhJaz1* was 4 times lower than that in the control plants at 3 hpi, then was slightly upregulated. *GhNOA* related to nitricoxide synthesis pathway, whose expression in silenced plants were significantly lower than that in control plants at 3, 6, 9 hpi. Similarly, *GhC4H1* was downregulated at 3, 6, and 9 hpi in silenced plants.

### 2.4. GhRPS6 Is Involved in Plant Resistance to V. dahliae by Regulating Hormone and Reactive Oxygen Species Levels

In order to explore how *GhRPS6* plays a role in plant disease resistance, the content of defense-related hormones and reactive oxygen species were measured at different development of roots. The results showed that the contents of Salicylic acid and Jasmonic acid in *GhRPS6*-silenced plants after inoculation was lower than that in the control plants. Particularly, a four-fold decreased of the contents of Jasmonic acid in *GhRPS6*-silenced plants compared with that in control plants. While the contents of Salicylic acid was lower in the silenced plants by 1 time compared with that in control plants (Figure 4a,b). The content of Fumaric acid in the *TRV:00* plants first rises, then descends at 6-48 h after inoculation. However, it showed an opposite trend in silenced plants. Especially, those were significantly lower than that in control plants after inoculation (Figure 4d). Reactive oxygen species (ROS) are considered as an important indicator of plant disease resistance. The results showed that the content of hydrogen peroxide (H_2_O_2_) in the root of silenced plants was lower than that in the root of control plants after inoculation. However, the content of NO in silenced and control plants changed as polymodal curve after inoculation (Figure 4e,f). Likewise, Diaminobenzidine (DAB) staining showed that the control plants had deeper staining and larger staining area, and demonstrated that the ROS levels were higher in leaves of control plant than that in leaves of silenced plants at 24 h post inoculation with Vd080 (Figure 4c). These results indicate that *GhRPS6* is involved in plant resistance to *V. dahliae* by regulating plant hormone signaling pathways and reactive oxygen species content.

### 2.5. Overexpression of GhRPS6 in Arabidopsis Enhances Plant Resistance

In order to further explore the resistance of *GhRPS6* to *V. dahliae*, *GhRPS6* was transferred into wild-type *A. thaliana* Col-0 by p*CAMBIA2300* vector, and the overexpression of *A. thaliana* was screened by 0.1% Kanamycin antibiotic and qPCR. The spore suspension was inoculated with T_3_ generation *A. thaliana* and Col-0. In MS plate and soil, the resistance of transgenic plants to *V. dahliae* was enhanced, and the disease index of wild-type and transgenic plants (OE:*GhRPS6*) were 48.65 and 26.76, respectively (Figure 5a). The results showed that the overexpression of *GhRPS6* in *A. thaliana* enhanced the resistance of plants to *V. dahliae* (Figure 5b).

In the early stage, LC-MS/MS analysis showed that Ser 237 and Ser 240 sites of GhRPS6 were phosphorylated upon infection of pathogen. In addition, the phosphorylation level of RPS6 in resistant cultivars was 1.5 times that of susceptible cultivars. To explore key phosphorylation sites, serines at 237 and 240 were mutated to phenylalanine, respectively and jointly [37,38]. The same method was used to screen the phosphorylated site inactivation mutant overexpressed plants by p*CAMBIA2300*. After inoculation, it was found that the plants with ser 237 mutation (OE:*GhRPS6^S237F^*) were small and weak. Similarly, the double mutant (OE:*GhRPS6^S237F/S240^^F^*) also found that the plants were small and weak, with reduced resistance (Figure 5c,d). The disease indexes of OE:*GhRPS6^S237F^*, OE:*GhRPS6^S240F^*, and OE:*GhRPS6^S237F/S240^^F^* were 65.1, 36.56, and 50.7, respectively (Figure 5d). It is speculated that ser 237 maybe is a key phosphorylation site, the loss of phosphorylation affects plant growth, making plants weak, and could weaken the resistance of GhRPS6 to *V.dahliae*.

### 2.6. Subcellular Localization

In order to obtain direct evidence of GhRPS6 protein localization, a *GhRPS6-GFP* vector was constructed, which was expressed instantaneously on the back of 4-week old tobacco leaves by agrobacterium injection. *CBF1-mCherry* is a vector known to be able to locate in the nucleus. It was observed after tobacco injection with *GhRPS6-GFP*. The results showed that GhRPS6 was located in tobacco nucleus and collocated with the red fluorescence signal. In order to explore the effect of *V. dahliae* on the localization of GhRPS6 protein, the position change of GhRPS6 was observed by soaking roots in the spore suspension of *V. dahliae* in tobacco after co-injection. The results showed that GhRPS6 fluorescence signal was stronger after inoculation. In order to explore the effect of phosphorylation sites on subcellular localization, the *GhRPS6^S237F^-GFP*, *GhRPS6^S^^240F^-GFP*, and *GhRPS6^S237^^/S240FF^-GFP* phosphorylation sites mutant vectors were constructed and transferred into *N. benthamiana* through *Agrobacterium* (Figure 6). It was found that phosphorylation inactivation and *V. dahliae* infection did not affect the localization of GhRPS6 on the nucleus.

## 3. Discussion

Verticillium wilt is the main disease in cotton production, which results in serious yield reduction in cotton field [35]. Breeding disease-resistant varieties is one of the most economical and effective control measures [4]. In recent years, more and more disease-resistant genes have been reported, such as the *G. hirsutum* TIR-NBS-LRR gene *GhDSC1*, which mediates resistance against Verticillium wilt [11]*,* while heterologous expression of the cotton NBS-LRR gene *GbaNA1* enhances Verticillium wilt resistance in *A. thaliana* [39], *GhWRKY70D13* regulates resistance to *V. dahliae* in cotton through the ethylene and Jasmonic acid signaling pathways [2], *GbMPK3* overexpression increases cotton sensitivity to *V. dahliae* by regulating Salicylic acid signaling [1], HyPRP1 performs a role in negatively regulating cotton resistance to V*. dahliae* via the thickening of cell walls and ROS accumulation [14], and an ethylene response-related factor, GbERF1-like, from *G. barbadense* improves resistance to *V. dahliae* via activating lignin synthesis [17]. The study of these genes has given people a deeper understanding of the molecular basis of resistance to *V. dahliae* in cotton, but the complexity of tetraploid cotton genome has hindered the cultivation of disease-resistant varieties. In the early stage, we found that GhRPS6 phosphorylation occurred in cotton after through proteomic analysis of cotton roots infestation by *V. dahliae*, and which phosphorylation level in resistant cultivars was significantly higher than that of susceptible cultivars [28]. On this basis, we put forward several questions, such as, is GhRPS6 related to cotton Verticillium wilt resistance? What immune responses are they involved in regulating? What is the key phosphorylation site of GhRPS6 protein? Does phosphorylation of GhRPS6 affect its function? Ribosomal biogenesis involves the manufacture and processing of ribosomal RNAs, the biosynthesis of ribosomal proteins, and the transport of ribosomal proteins to the nucleolus to assemble the pre-ribosomal particles [40,41,42,43,44,45,46,47,48]. Existing literature shows that ribosomal proteins are involved in various physiological functions of cells and have been reported to regulate plant growth [41]. For example, ribosomal protein NtRPL17 interacts with kinesin-12 family protein NtKRP and functions in the regulation of embryo/seed size and radicle growth [49], cytoplasmic ribosomal protein L14B is essential for fertilization in *Arabidopsis* [50], Dek44Maize encodes mitochondrial Ribosomal Protein L9 and is required for seed development [51], the essential chloroplast ribosomal protein uL15c interacts with the chloroplast RNA helicase ISE2 and affects intercellular trafficking through plasmodesmata [52], down-regulation of specific plastid ribosomal proteins suppresses thf1 leaf variegation, implying a role of THF1 in plastid gene expression [53], Salicylic acid-related cotton (*G. arboreum*) ribosomal protein GaRPL18 contributes to resistance to *V. dahliae* [16]. TOR and RPS6 enhance protein translation in *A. thaliana* seedlings by transmitting light signals [54], ribosomal protein S6 may be involved in the regulation of rRNA genes through epigenetic changes in *A. thaliana* [55]. In this study, the leaf wilting and yellowing in *TRV:GhRPS6* plants were more serious, and vascular bundle browning was significantly changed. Which showed that GhRPS6 silencing reduced the resistance of cotton to *V. dahliae*, indicating that GhRPS6 played a positive regulatory role in plant resistance to pathogenic fungus infection. Due to the complexity of heterotetraploid genomes (subgroup A and subgroup D) in upland cotton (*G. hirsutum*) [56], we first conducted phylogenetic analysis of four cotton cultivars, *G. hirsutum* (AADD), *G. barbadense*(AADD), *G. arboreum* (donor subgroup A) and *G. raimondii* (donor subgroup D), and *A. thaliana*. We identified 6 *RPS6* genes in each of *A. thaliana* and *G. arboreum*, 12 *RPS6* genes in *G. raimondii*, 14 *RPS6* genes in *G. barbadense*, and 15 *RPS6* genes in *G. hirsutum* (Figure 1). Chromosome mapping showed that among the 15 *GhRPS6* genes in *G. hirsutum*, 10 were distributed in subgroup D and 5 were distributed in subgroup A (Appendix A). These results indicated that *RPS6* gene loss occurred in *G. hirsutum* and *G. barbadense*, which was consistent with a higher gene loss rate in allotetraploid cotton than in both diploid species [57]. It also showed that the ancestors of *G. raimondii* provided more *RPS6* gene donors to *G. hirsutum* than *G. arboreum*. Transcriptome analysis showed that *GhRPS6* (Gh_D13G0519.1) expression was very low in different tissues, and decreased after induction of highly pathogenic deciduous strain Vd991 and attenuated strain Vd07083. After VIGS silencing *GhRPS6*(Gh_D13G0519.1), cotton plants had more severe disease and increased disease index (Figure 2). This suggested that GhRPS6 was more involved in cotton resistance to *V. dahliae* through the modification level (protein phosphorylation). After being infected by pathogenic fungi, plants will trigger a series of defense reactions to resist the infection of pathogenic fungi [58], including the outbreak of reactive oxygen species [14], xylem thickening [17], callose accumulation, and so on [59]. In this study, it was found that silencing *GhRPS6* affected the defense response of cotton against *V. dahliae* infection, leading to increased fungal colonization (Figure 2). Plant hormone is an important signal factor in plant defense response [60]. Salicylic acid and Jasmonic acid regulate the expression of downstream genes by regulating cascade signaling pathways to resist pathogen fungal infection [61]. For example *GbMPK3* overexpression increases cotton sensitivity to *V. dahliae* by regulating Salicylic acid signaling [1], cotton *WATs* modulate SA biosynthesis and local lignin deposition participating in plant resistance against *V. dahliae* [62], tomato *SlMAPK3* induces plant disease resistance by regulating plant hormone content [63]. This study found that *GhRPS6* reduced the contents of Salicylic acid and Jasmonic acid after silencing (Figure 4). After the pathogen fungus infects the plant, the plant cells produce reactive oxygen species (ROS) to remove the exogenous invaders [64]. DAB staining found that the leaves of control plants had deeper staining and larger staining area, and demonstrated that the ROS levels were higher in leaves of control plants than that in the leaves of silenced plants (Figure 4). The determination of H_2_O_2_ content confirmed this conclusion (Figure 4f). H_2_O_2_ induces NO production, however, nitric oxide synthesis pathways include NOS pathway, NR pathway, and non-enzymatic pathway, which also lead to the dramatic fluctuation of NO content within one hour. Chalcone Isomerase (CHI) is a rate-limiting enzyme in the synthesis pathway of flavonoids, which is related to plant disease resistance [65,66]. Phenylalanine Ammonia lyase (PAL) is closely related to the synthesis of plant antitoxins and phenolic compounds [67]. It is a marker gene of plant disease resistance. Prophenoloxidase (PPO) catalyzes the synthesis of quinones, which prevent pathogens from reproducing in plants [68]. Jasmonate ZIM (Zinc-finger Inflorescence meristem) domain (JAZ) is a transcriptional inhibitor of the Jasmonic-acid signaling pathway in plants, silencing Jasmonic-acid-induced genes [69,70,71]. Nitricoxide associated factor (NOA) is a gene related to nitricoxide synthesis pathway, which participates in plant disease resistance by regulating NO content [72,73]. Cinnamate 4-hydroxylase (C4H1) is a key enzyme in lignin synthesis that plays a role in disease resistance in plants by regulating lignin content [74,75]. The results showed that the expression of disease-resistant genes in *GhRPS6* silencing plants was significantly decreased after inoculation. It was jointly proven that *TRV: GhRPS6* reduced the cotton resistance to *V. dahliae* due to the reduced lignin synthesis, reduced ROS outbreak, and decreased fungal colonization inhibition ability.

Studies have shown that epigenetics, especially protein phosphorylation, play important roles in plant defense [76,77,78]. For example, abiotic stress antagonized the rice defense pathway through OsMPK6 tyrosine dephosphorylation [76]. In *Arabidopsis*, WHIRLY1 phosphorylation by CIPK14 alters its localization and dual functions [79]. The results showed that the overexpression of *GhRPS6* enhanced the resistance of transgenic *A. thaliana* to *V. dahliae,* and the non-phosphorylation of Ser-237 affected the disease resistance of the overexpressed *Arabidopsis thaliana* (Figure 4). These results indicated that the phosphorylation and inactivation of *GhRPS6* affected the resistance of *A. thaliana* to *V. dahliae*. The biogenesis of eukaryotic ribosomes mainly occurs in the specific subnuclear region of nucleolus, and involves the coordinated assembly of ribosomal RNA and ribosomal proteins [80,81]. Identification of amino acid sequences that mediate ribosomal protein nucleolus localization may provide important clues for understanding the early steps of ribosomal biogenesis [81]. For example, the localization of human ribosomal protein S17 in the nucleoli may be related to Diamond-Blackfan Anemia (DBA) [80]. GhRPS6 is a ribosome S6 protein, and nuclear localization sequence prediction found that amino acid sequence contains five NLS, and transmembrane domain prediction did not exist. Tobacco localization found that the protein was localized in the nucleus. *GhRPS6^S237F^-GFP, GhRPS6^S^^240F^-GFP*, and *GhRPS6^S237^^/S240FF^-GFP* were expressed instantaneously in *N. benthamiana*, and it was found that the infection of *V. dahliae* had no effect on the localization of proteins in *N. benthamiana* cells. In conclusion, phosphorylation of GhRPS6 protein does not affect its cell localization. However, the phosphorylation level of GhRPS6 protein is related to its function, in particular the mutation of GhRPS6 at site Ser237 with the phosphorylation compromised the plant resistance to *V. dahliae*.

In conclusion, this study explored the relationship between *GhRPS6* and resistance to Verticillium wilt in cotton, and found that the transcription level of *GhRPS6* in tissues was low, and the expression of *GhRPS6* decreased after inoculation. When the *GhRPS6* gene was silenced in cotton, the plant resistance was weakened, and the overexpression of *GhRPS6* in *A.thaliana* increased the plant resistance, while the phosphorylation significantly affected the disease resistance of *A.thaliana* plants. Subcellular localization indicated that GhRPS6 was localized in the nucleus, and inoculation and phosphorylation inactivation mutations had no effect on nuclear localization, indicating that GhRPS6 did not affect plant resistance through expression level and localization, but regulated plant resistance through phosphorylation. We propose that GhRPS6 is a potential molecular target for improving resistance to Verticillium wilt in cotton.

## 4. Materials and Methods

### 4.1. Plant Materials, Fungal Strain, and Growth Conditions

The seeds of upland cotton (*G. hirsutum*) cultivar ‘Zhongzhimian 2′, which are highly resistant to *V. dahliae* [82], were provided by the Institute of Cotton Research of Chinese Academy of Agricultural Sciences. *G. hirsutum* and *N. benthamiana* were sown in soil and the seedlings were grown in a greenhouse under 8 h/16 h dark/light cycle and at 23 °C (dark)/28 °C (light) with a relative humidity of 60%.

*A. thaliana* seeds were soaked in 75% ethanol for 30S, washed with sterile water, and soaked in 3% sodium hypochlorite for 3 min, then washed with sterile water for three times and inoculated in MS solid medium without antibiotics. After a week (four leaves) in a light incubator (24 °C-16 h/20 °C-8 h), transfer in pots with potting soil including 40% vermiculite, greenhouse at temperatures of 24 °C during the day and 20 °C at night, 60–70% relative humidity, and under a 16/8 light/dark photoperiod.

In this study, the deciduous strain Vd080 with strong pathogenicity was selected and stored in the Institute of Cotton Research of Chinese Academy of Agricultural Sciences [83]. It was grown on PDA medium (potato dextrose agar) at 25 °C. The mycelia were collected and cultured in liquid Czapek’s medium as described previously. The final concentration of the spore suspension was adjusted to 1 × 10^7^ conidia/mL with sterile water.

### 4.2. Gene Cloning and Sequence Analysis

*GhRPS6* CDS sequences were searched in cotton database (https://cottonfgd.org/profiles/transcript/Gh_D13G0519.1/ (accessed on 1 December 2020)) and gene specific primers were designed using Primer5.0 (Appendix A). The coding sequence of *GhRPS6* (Gh_D13G0519) was amplified from roots of ‘Zhongzhimian 2′ using high fidelity enzyme. The total RNA was extracted using RNAprep Pure Plant Kit (TIANGEN) according to the protocol provided by the manufacturer. *TransScript^®^* All-in-One first-strand cDNA Synthesis SuperMix for qPCR (One-Step gDNA Removal) reverse transcription kit was used to synthesize cDNA. The amplified product was cloned into the p*EASY*^®^-Blunt Cloning Kit vector, and confirmed by sequencing. The gene structure of *GhRPS6* (sequence cloned from the root of ‘Zhongzhimian 2′) was analyzed by online analysis software (http://gsds.cbi.pku.edu.cn/ (accessed on 1 December 2020)) [84]; others were used to predict the phosphorylation sites (http://www.cbs.dtu.dk/services/NetPhos/ (accessed on 1 December 2020)) of the proteins and the hydrophilicity/hydrophobicity (https://web.expasy.org/protscale/ (accessed on 1 December 2020)) of GhRPS6 proteins [85], using the CDD NCBI database (http://www.ncbi.nlm.nih.gov/Structure/cdd/wrpsb.cgi (accessed on 1 December 2020)) predictive coding protein structure existing in the field [86]; using the analysis method of TMHMM (http://www.cbs.dtu.dk/services/TMHMM (accessed on 1 December 2020)) proteins across the membrane [87]. The secondary structure (http://bioinf.cs.ucl.ac.uk/psipred/ (accessed on 1 December 2020)) and tertiary structure (https://www.swissmodel.expasy.org/ (accessed on 1 December 2020)) of GhRPS6 are analyzed by online network analysis software [88].

### 4.3. Phylogenetic Analysis

Proteomic data for cotton and Arabidopsis were obtained from CottonFGD (https://cottonfgd.org/ (accessed on 1 December 2020)) and TAIR10 (http://www.arabidopsis.org/ (accessed on 1 December 2020)), respectively. Multisequence alignment of all RPS6 proteins was performed using ClustalX with default parameters [32]. A phylogenetic tree of the inferred RPS6 amino acid sequence was constructed using a neighbor-joining algorithm with default parameters.

### 4.4. Gene Structure Analysis and Conserved Motif Identification

The Gene Structure Display Server Program (http://gsds.cbi.pku.edu.cn/ (accessed on 1 December 2020)), according to the total length of the genome sequence and the corresponding coding sequence, mapped the *GhRPS6* gene exon-contains substructure [84]. The MEME program was used to identify the conserved domain in GhRPS6 protein [56]. MCScanX was used to identify genome collinearity and tandem repeats under default parameters, and Circosv.0.69 was used to show collinearity [32].

### 4.5. GhRPS6 Genes Silenced and Fungal Pathogen Inoculation

The vectors of tobacco rattle virus (TRV) were used for VIGS to investigate the function of *GhRPS6* in response to *V. dahliae* infection [89]. The fragment *GhRPS6* was amplified by PCR using ‘Zhongzhimian 2′ cDNA with the GhRPS6–VIGS-F/R primers, and then cloned into the *TRV:00* plasmid at the XbaI-SacI site using the T4 DNA Ligase (TaKaRa). The connection product was transferred to DH5α by heat shock method and cultured overnight in LB plate at 37 °C. After sequencing correctly, the TRV vectors were transformed into *Agrobacterium tumefaciens* GV3101 chemically. The strains were mixed with 30% sterile glycerin 1:1 and stored at −80 °C. *A. tumefaciens* was cultured in LB liquid medium containing kanamycin and rifampicin at 28 °C 180 rpm until OD_600_ = 1.0. Centrifuged at 5000 rpm for 10 min to collect the bacteria, the MMA liquid (10 mM/L MgCl, 10 mM/L MES, 200 μM/L AS) was resuspended, and OD_600_ = 0.8 was adjusted after washing for three times. *Agrobacterium* harboring the empty vector of p*YL-156* or one of its derivatives (p*YL-156-GhRPS6*, p*YL-156-GhPDS*) were mixed with an equal volume of *Agrobacterium* harboring p*YL-192*. TRV vectors injected into two fully expanded cotyledons of 10-day-old upland cotton seedlings by Agroinfiltration method [89]. The culture was normal after dark treatment for 24 h. After *GhPDS* was silenced in cotton, albino phenotype appeared in leaves and stems as a positive control (about 10 day). *TRV:00* and *TRV:GhRPS6* plants were used to detect the expression level of *GhRPS6* in order to detect the silencing efficiency.

After the albino phenotype appeared in the positive control group for about 10 days, the successfully silenced plants were inoculate the *V. dahliae*. Vd080 with conidia suspension (1×10^7^ conidia/mL) by the root dipping method for 10 min and then replanted [89].

### 4.6. Analysis of ROS, Callose, and the Xylem

The generation and accumulation of reactive oxygen species in cotton leaves 24 h after inoculation were detected by 3,3 ‘-diaminobenzidine (DAB) staining [90]. A true leaf of *TRV:00* and TRV: *GhRPS6* plants was randomly selected, the surface was rinsed with distilled water and the water was sucked dry. DAB staining solution was dyed at room temperature and away from light for 8 h, and 95% ethanol was added to boil water bath to remove the chlorophyll. After the chlorophyll was completely removed, the leaves were immersed in 70% glycerin for microscope observation and photography. Each experiment was repeated three times.

Five cotton seedlings were randomly selected 48 h after inoculation, and phloroglucinol staining was used to observe the xylem discoloration [62]. The same parts of the stem of *TRV:00* and *TRV:GhRPS6* plants were sliced and stained with 10% phloxol solution (dissolved in anhydroethanol) for 2 min and then added with concentrated sulfuric acid rapidly, which was observed under a microscope and photographed. Three independent biological and technical repeats were performed.

The leaves of five cotton seedlings of *TRV:00* and *TRV:GhRPS6* were randomly selected 48 h post inoculated with bacteria, respectively. The leaves were soaked in ethanol and acetic acid with a volume ratio of 3:1 for three hours to remove the chlorophyll, and then soaked in 70% and 50% ethanol for three hours, respectively, and then soaked overnight in distilled water. The leaves were rinsed the next day and then treated with 10% sodium hydroxide (NaOH) for two hours, washed four times, then cultured in 0.01% aniline blue in dark for three hours. The callose density was observed under a fluorescent microscope. Three independent biological and technical repeats were performed.

After 25 days of inoculation with *Verticillium* of cotton, 7 plants of *TRV:00* and 7 plants of *TRV:GhRPS6* were randomly selected. One cm long fragments from the first stem node were collected and disinfected on a clean bench (75% ethanol for 3 times, 3% sodium hypochlorite for 3 min), and sterile distilled water then placed on a PDA plate, and cultured at 25 °C for 7 days for observation. Three independent biological and technical repeats were performed.

Five strains of *TRV:00* and *TRV: GhRPS6* were randomly selected 25 days after inoculation with Verticillium wilt of cotton, and the first true leaf was taken for use. 10 mL lactic acid, 10 mL glycerol, 10 g phenol, 10 mg trypan blue, and 10 mL distilled water were mixed with trypan blue dye solution. The leaves were immersed in trypan blue dye solution, boiled in water for two minutes, and dyed overnight at room temperature after natural cooling. The decolorization solution was changed every 24 h for three days at 2.5 g/mL chloral hydrate. Finally, the cell death was observed under stereomicrograph and photographed. The experiment was repeated three times.

The middle and lower sections of inoculated cotton stem were divided into longitudinal and oblique sections, respectively, to observe the browning of xylem.

### 4.7. Measurements of H_2_O_2_, No, JA and SA

Three *TRV:00* or *TRV:GhRPS6* plants were randomly selected at corresponding time points after inoculation, and stored at −80 °C after liquid nitrogen treatment. Quantitative Assay Kit (Nanjing Jiancheng, Beijing, China) was used for the determination of hydrogen peroxide (H_2_O_2_) and nitric oxide (NO), while Jasmonic acid (JA) and Salicylic acid (SA) were tested by a high performance liquid chromatography (HPLC) system (Agilent 1260). Three independent biological and technical repeats were performed.

### 4.8. RT-qPCR Analysis

To monitor the expression levels of related resistance genes, cotton plants with *TRV:00* and *TRV:GhRPS6* were randomly collected at 1, 3, 6, 12, and 48 h post inoculation, and stored at −80 °C after liquid nitrogen treatment. Mechanically milled plant leaves are powdered. The total RNA was extracted using RNAprep Pure Plant Kit (TIANGEN) according to the protocol provided by the manufacturer. *TransScript^®^* All-in-One first-strand cDNA Synthesis SuperMix for qPCR (One-Step gDNA Removal) reverse transcription kit was used to synthesize cDNA, and *TransStart*^®^ Tip Green qPCR SuperMix kit was used for qPCR. RT-PCR and qPCR circulations and reaction procedures were carried out in accordance with the instructions. Cotton *ubiquitin 7* (*UB7*) genes were amplified as reference gene, and the relative fold changes of the target genes were calculated using the 2^−ΔΔCt^ method [89]. The primers used in this study are listed in Appendix A. Three independent biological and technical repeats were performed.

### 4.9. Plant Disease Resistance Assess

When the leaves of the plant withered and turned yellow, the disease plant was graded the disease index was calculated according to the formula [89]:
Disease index = [(∑level *n* × number of diseased plants at level *n*)/(total checked plants × 4)] × 100.


All experiments were repeated three times, with more than 80 plants planted each time.

### 4.10. Construction and Screening of Overexpressed Arabidopsis thaliana

The ORF of *GhRPS6*, *GhRPS6^S237F^*, *GhRPS6^S240F^*, and *GhRPS6^S237F/S240F^* was cloned under control of the 35S promoter in the plant expression vector p*CAMBIA2300*. Then, which were introduced into the *A. tumefaciens* strain GV3101. *Agrobacteria* containing *35S-GhRPS6* (*GhRPS6^S237F^*, *GhRPS6^S240F^*, and *GhRPS6^S237F/S240F^*) vector were cultured at 28 °C/180 rpm to OD_600_ = 2.0, collected by centrifugation, and resuspended with sterilized infection solution (MS medium 4.3 g/L, sucrose 50 g/L, Silwet77 200 μL/L). Transformation of *A. thaliana* was performed using the floral-dip method [89]. The harvested T_0_-T_3_ generation seeds were disinfected and planted in MS medium containing 50 mg⋅L^-1^ Kan. T_3_ lines with the transgene and the correct segregation ratio were identified with RT-qPCR analysis. The primers used in this study are listed in Appendix A. 

### 4.11. Subcellular Localization

When the agrobacterium containing the vector *35S-GhRPS6-GFP* (and *35S-GhRPS6^S237F^-GFP*, *35S-GhRPS6^S240F^-GFP*, *35S-GhRPS6^S237F-GFP /S240F^-GFP*) were cultured to OD_600_ = 1.0, the bacteria were collected by centrifugation, and the heavy suspension was diluted to OD_600_ = 0.8. At the same time, *Agrobacterium* containing plasmid *CBF1-mCherry* was cultured in the same way, collected, re-selected to the same concentration and mixed in the same volume, then the bacteria were left for 2 h at room temperature. The tobacco leaves were co-injection into the back of 4-week old tobacco leaves. The dark culture was moved to the normal culture environment for 24 h, and the leaves were taken for glass slides 48 h later, and observed by laser scanning confocal microscope.

## Figures and Tables

**Figure 1 ijms-22-01795-f001:**
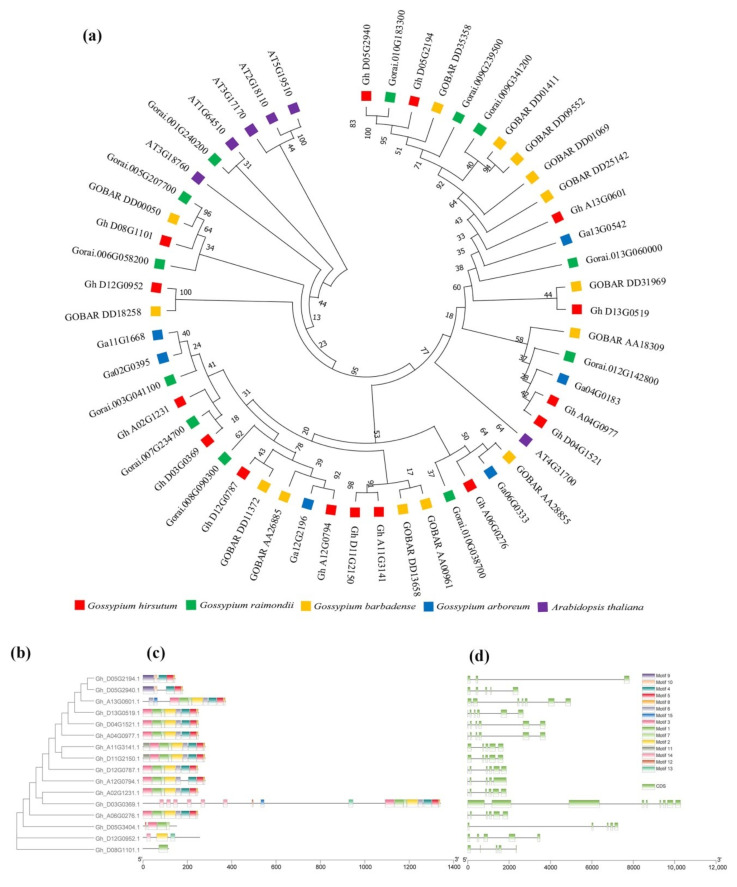
Phylogenetic tree analysis phylogenetic tree. (**a**) Rootless phylogenetic trees were constructed using the protein sequences from *Arabidopsis thaliana, Gossypium hirsutum, Gossypium raimondii, Gossypium arboreum,* and *Gossypium barbadense*; (**b**) the phylogenetic tree of the GhRPS6 gene in *Gossypium hirsutum* (left), was constructed using the neighbor-joining (NJ) method in MEGA-X and has 1000 copies of the bootstrapping program; (**c**) motif analysis of *GhRPS6* genes. Identify all patterns with the MEME software (https://meme-suite.org/meme/tools/meme (accessed on 1 December 2020)). The length of each pattern is shown in proportion; (**d**) analysis of *GhRPS6* gene structure. Use the Gene Structure Display Server 2.0 to draw the gene structure diagram with the scale shown at the bottom.

**Figure 2 ijms-22-01795-f002:**
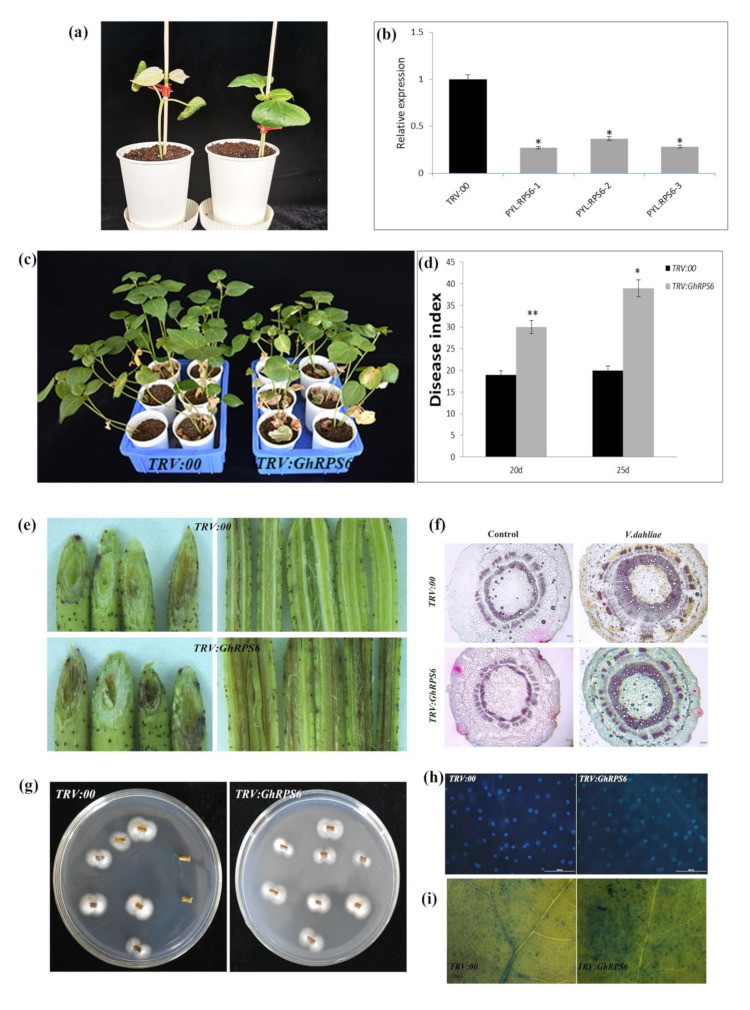
*GhRPS6* gene silencing reduced the resistance of cotton to *V. dahliae*. (**a**) Albino phenotype of *GhPDS* appeared 10 days after injection; (**b**) *GhRPS6* gene silence efficiency detection; (**c**) pathogenetic phenotypes of control and *GhRPS6*-silenced plants; (**d**) disease index at 20dpi and 25 dpi; (**e**) stem vascular bundle Browning; (**f**) stem slices were stained with vascular bundles, the scale is 100 microns; (**g**) the recovery assay of *V. dahliae*; (**h**) callose deposition in leaves of the control and *GhRPS6*–silenced plants at 21 days after Vd080 inoculation, the scale is 200 microns; (**i**) trypan blue experiment. Error bars represent the standard deviation of three biological replicates. Asterisks indicate statistically significant differences as determined by Student’s t test (* *p* < 0.05; ** *p* < 0.001).

**Figure 3 ijms-22-01795-f003:**
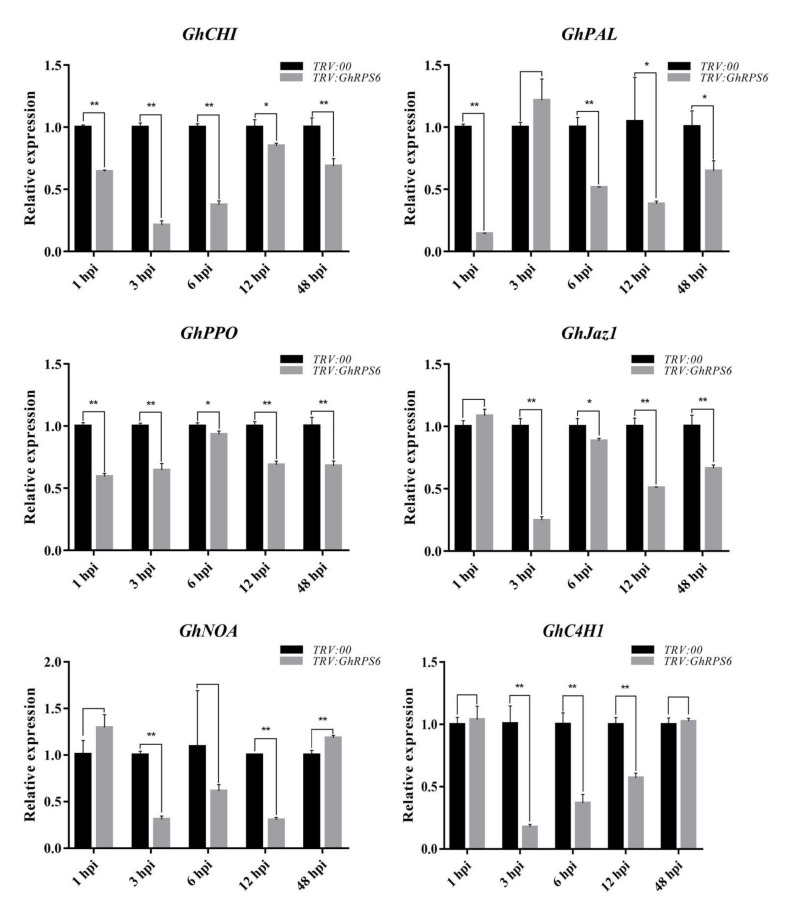
Expression levels of disease resistance-related genes in leaves of *GhRPS6*–silenced and control cotton plants. Error bars represent the standard deviation of three biological replicates. Asterisks indicate statistically significant differences, as determined by Student’s *t*-test (* *p* < 0.05; ** *p* < 0.001).

**Figure 4 ijms-22-01795-f004:**
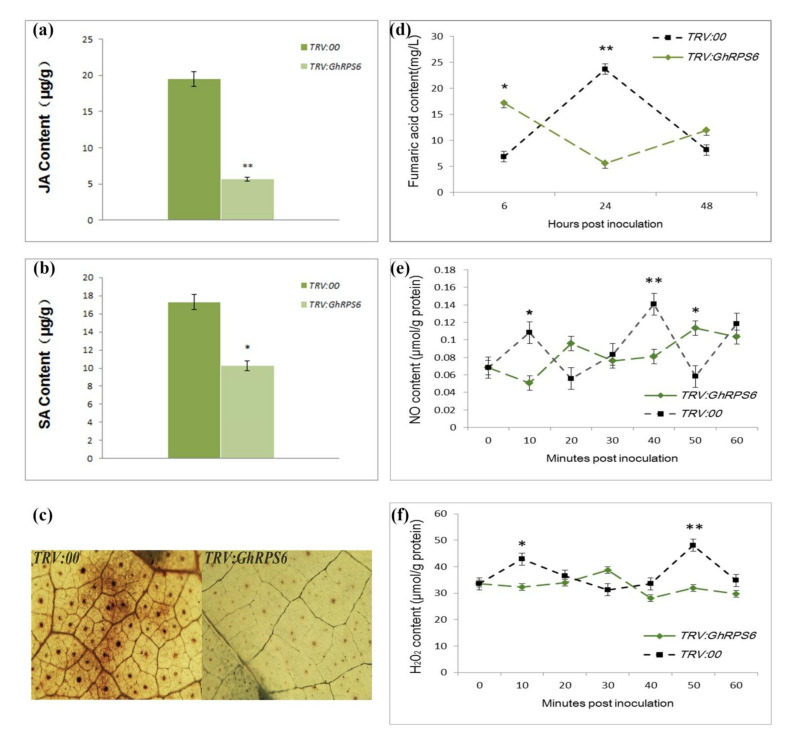
The accumulation of Jasmonic aci, Salicylic acid, Fumaric acid, H_2_O_2_, and NO in *GhRPS6* silenced and control plants. (**a**,**b**) Jasmonic acid and Salicylic acid content in silenced plants and control plants after infestation; (**c**) reactive oxygen species (ROS) occurred in leaves of silencing plants and control plants at 24 h after inoculation, ROS accumulation was captured by the microscopy with 10×amplification under the stereomicroscope; (**d**) the content of Fumaric acid in the roots of silenced plants and control plants inoculated with *V. dahliae* for 6-48h; (**e**,**f**) the content of hydrogen peroxide (H_2_O_2_) and nitric oxide (NO) in the roots of silenced plants and control inoculated with *V. dahliae for* 1 h (* *p* < 0.05; ** *p* < 0.001).

**Figure 5 ijms-22-01795-f005:**
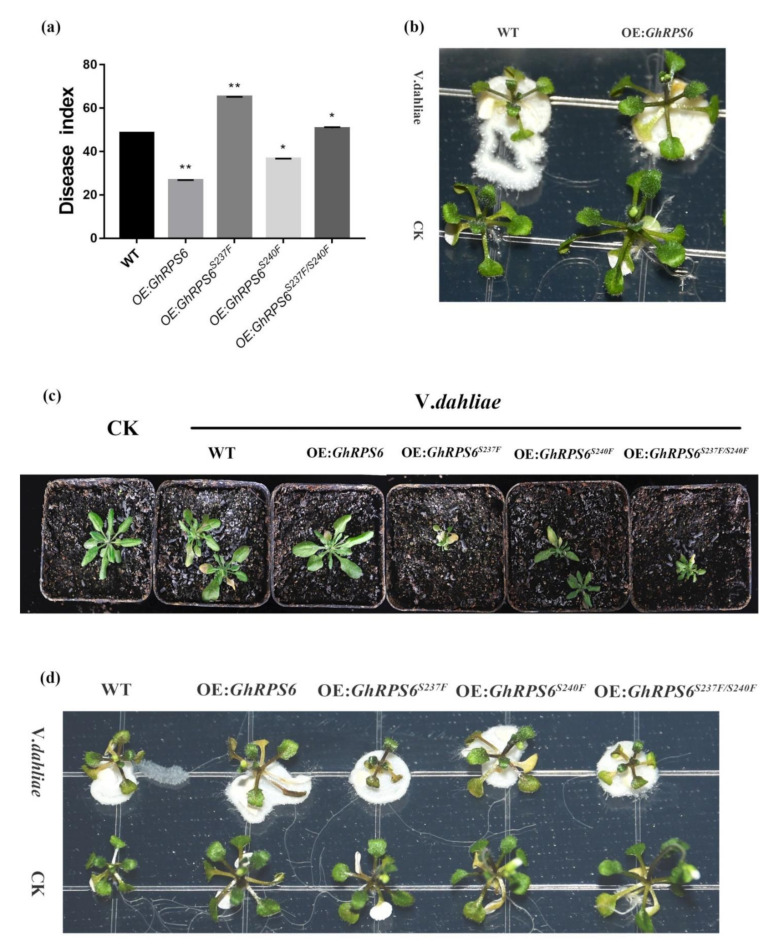
The overexpression of *GhRPS6* in *A. thaliana* enhanced the resistance of the overexpressed plants to *V. dahliae*. (**a**) Disease index, error bars represent the standard deviation of three biological replicates. Asterisks indicate statistically significant differences as determined by Student’s t test (* *p* < 0.05; ** *p* < 0.001); (**b**) WT, OE:*GhRPS6* inoculation in MS plate for two days; (**c**) WT, OE:*GhRPS6*, OE:*GhRPS6^S237F^*, OE:*GhRPS6^S240F^*, OE:*GhRPS6^S237F/S240F^* inoculation in soil for 20 days; (**d**) WT, OE:*GhRPS6*, OE:*GhRPS6^S237F^*, OE:*GhRPS6^S240F^*, OE:*GhRPS6^S237F/S240F^* inoculation in MS plate for two days.

**Figure 6 ijms-22-01795-f006:**
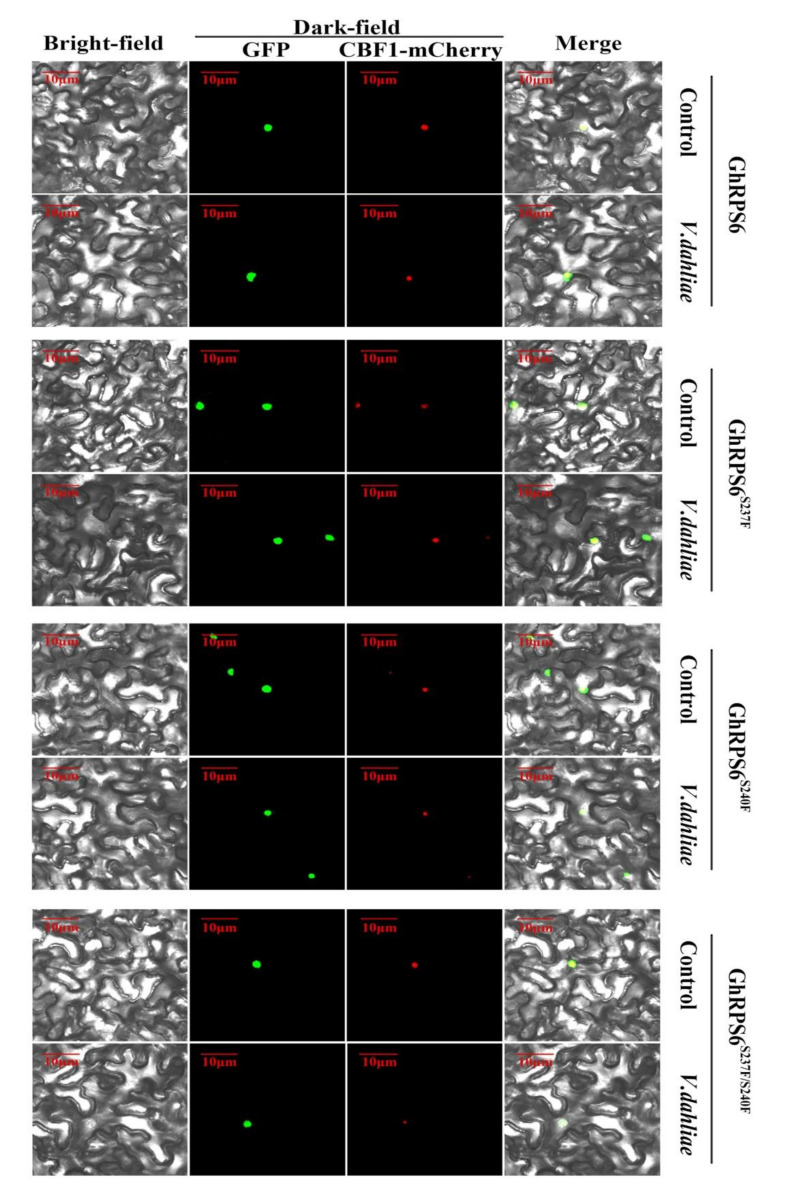
Subcellular localization of GhRPS6, GhRPS6^S237F^, GhRPS6^S2^^40F^, and GhRPS6^S237F^^/S240F^. CBF1 is a protein that has been reported to be located in the nucleus. *GhRPS6*, *GhRPS6^S237F^*, *GhRPS6^S2^^40F^*, and *GhRPS6^S237F^^/S2^^40F^* are connected to the GFP label, and both of them are located in the nucleus in *N. benthamiana*.

## Data Availability

Not applicable.

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
