# Peer review of "Genome-Wide Analysis of Ribosomal Protein GhRPS6 and Its Role in Cotton Verticillium Wilt Resistance"

_ijms, 2021, doi:10.3390/ijms22041795_

Round 1

Reviewer 1 Report

The manuscript deals with the identification of a new protein involved in the response of cotton to Verticillium dahliae. VIGS of this gene resulted in increased disease, based on biological and molecular data, suggesting its involvement in the resistance to V. dahliae. On the other hand, data on transgenic Arabidopsis and phosphorylation are not convincing at all, and conclusions arisen from that might be biased.
Then, the manuscript has several editorial concerns. The title does not reflect the research or the main finding. In the Introduction, the resistance of cotton (different species) to V. dahliae should be better elaborated, while the part dealing with the resistance mechanisms (L49-65) could be removed because it is not pertinent. Discussion is very poor and basically a repetition of the results: the only good part is that dealing with phosphorylation; in this section, the authors should discuss their results with the literature to understand the strength and weakness of the research. Methods are lacking many details for all experiments (including growth conditions, experimental design, replicates, procedures, etc.); also, citations to previous works could be used to avoid repetition of method description. The previous work on proteomic should be better presented and described, and the present work should be better circumstantiated.
English needs an accurate revision.
In the present form, the article should be rejected. However, maybe that significant improvement and/or additional experiments on phosphorylation could make the manuscript acceptable for publication in the International Journal of Molecular Sciences.

Author Response

Reviewer #1: The manuscript deals with the identification of a new protein involved in the response of cotton to Verticillium dahliae. VIGS of this gene resulted in increased disease, based on biological and molecular data, suggesting its involvement in the resistance to V. dahliae. On the other hand, data on transgenic Arabidopsis and phosphorylation are not convincing at all, and conclusions arisen from that might be biased.
Then, the manuscript has several editorial concerns.:

1. The title does not reflect the research or the main finding.

Our response: Thanks for your suggestion. We have changed the title.

  1. In the Introduction, the resistance of cotton (different species) to V. dahliae should be better elaborated, while the part dealing with the resistance mechanisms (L49-65) could be removed because it is not pertinent.

Our response: Thanks for your suggestion. We have revised the introduction according to your suggestion and added the description of different disease-resistant varieties of cotton.

  1. Discussion is very poor and basically a repetition of the results: the only good part is that dealing with phosphorylation; in this section, the authors should discuss their results with the literature to understand the strength and weakness of the research.

Our response: Thanks for your suggestion. We have modified the discussion part according to your suggestion, added the comparison between the results of this study and previous experiments, and deleted the repetition of the results.

  1. Methods are lacking many details for all experiments (including growth conditions, experimental design, replicates, procedures, etc.); also, citations to previous works could be used to avoid repetition of method description. The previous work on proteomic should be better presented and described, and the present work should be better circumstantiated.

Our response: Thanks for your suggestion. We have modified the materials and methods according to your suggestions, and added references of previous experimental methods.

  1. English needs an accurate revision.

Our response: Thanks for your suggestion. We have checked the whole manuscript carefully for the grammar ortypo error, and reviewed the manuscripts in standard English. In addition, this manuscript has been edited by a native English professional with science background. We hope that it is acceptable for publish.

  1. In the present form, the article should be rejected. However, maybe that significant improvement and/or additional experiments on phosphorylation could make the manuscript acceptable for publication in the International Journal of Molecular Sciences.

Our response: Thank you very much for your advice. The article has been major revisions, included Introduction, Results, Discussion and Materials and Methods. For the phosphorylation, we only demonstrated that seri-237 may be compromised the plant resistance to V. dahliae. It was indeed a sorry sight. We have been also looking for more convincing evidence to better understand the role of phosphorylation.

Reviewer 2 Report

In this article, Zhu et al, tried to describe GhRPS6 role in disease resistance in cotton. However, it seems that the study was conducted without focusing on a single objective and clear hypothesis. The author has done most bioinformatics analyses that do not seem to be relevant to the study. The gene was studied in cotton, the OX line was studied in Arabidopsis while localization was studied in tobacco. If OX line generation was difficult in cotton, then why Arabidopsis was not used for cellular localization. Some critical shortcomings need to be addressed. Based on this assessment, I do not recommend the paper for publication in its current form.  

There are serious grammatical, spelling, and sentence structure problems. I understand that English is not the first language of authors, however, the way the manuscript has language mistakes, the interpretation might be taken wrong by a potential reader.  The reviewer does not expect such a lack of seriousness and caution towards basic mistakes even in the title. This is the responsibility of all the co-authors to check the manuscript thoroughly before submission.

Title: A ribosomal protein GhRPS6 was dentified??

Line 129: showed that The, T in the should be small

Line 161: GhRPS6 silences reduced the … give a wrong meaning.

Line 169: proved to be no problem??

Line 174: incidence of silent plants was more higher, should be changed

Line 197: hr should be h

Line 202: And then, whose was increased??

Line 396 washed should be changed to rinsed.

Line 399: there should be no space between the number and degree sign

Line 404: the number should be written incorrect notation.

Line 511: promoterin??

Line 516: Arabidopsisplants??

Other comments

Line 103: in the early proteomic analysis? Which analysis, this is the first result?

The genotyping results of both OX and silenced plants should be given, as a supplementary table.

How NO was measured was not clear. NO being highly reactive combines with other molecules to make SNOs or other RNS, it is therefore strange that how NO was exactly measured.

Results section 2.1. how these analyses are related? Why the phylogenetic tree was constructed? Does the gene expression level of GhRPS6 has checked in cotton and its orthologous plants?

If the gene was cloned from the resistance cultivar? was the level of expression checked? The gene might already be overexpressed in the tolerant cultivar if it is involved in disease resistance, then why a 35S pro line was produced.

Figure 2 A: the author should also show mock-treated plants for comparison.

Figure 2, callose deposition and trypan blue staining needs to be repeated

Figure 3 has no labels.

Figure 4 Pannel A and B are the both JA?

Figure 4: H2O2 induces NO production, then after 50 minutes post-inoculation, the TRV00 showed the highest H2O2 contents, while at the same time the same genotype showed reduced NO content. And if the NO contents data is correct, and If NOA is involved in NO synthesis, we would assume that high NO content in TRV00 after 60 minutes might be because of induced NOA expression to make more NO, however, the NOA expression in TRV00 at 1 hpi shows the opposite results. Can the author explains why?

The NO contents in TRV00 were high at 10 m reduced at 20, again raised at 30 and 40 and then reduced at 50. Can the author explain this zigzag nature of NO content in TRV00?

Does the c panel show DAB staining? The staining shows a clear difference in H2O2 level among the two plants after 24 h, however, it is not reflected in the data

There is no coherence in the parameters. The NO, H2O2, contents were measured in a time interval of minutes with a maximum for 1 h, SA and JA contents were measured at 24 h, while the gene expression was studied up to 48 h. Hussain et al., 2016. doi: 10.3389/fpls.2016.00975. eCollection 2016 showed global changes in gene expression after 6 h of NO donor application. what was the logic to study NO and H2O2 contents that early?

Author Response

Reviewer #2: In this article, Zhu et al, tried to describe GhRPS6 role in disease resistance in cotton. However, it seems that the study was conducted without focusing on a single objective and clear hypothesis. The author has done most bioinformatics analyses that do not seem to be relevant to the study. The gene was studied in cotton, the OX line was studied in Arabidopsis while localization was studied in tobacco. If OX line generation was difficult in cotton, then why Arabidopsis was not used for cellular localization. Some critical shortcomings need to be addressed. Based on this assessment, I do not recommend the paper for publication in its current form. 

There are serious grammatical, spelling, and sentence structure problems. I understand that English is not the first language of authors, however, the way the manuscript has language mistakes, the interpretation might be taken wrong by a potential reader.  The reviewer does not expect such a lack of seriousness and caution towards basic mistakes even in the title. This is the responsibility of all the co-authors to check the manuscript thoroughly before submission.

Our response: Thank you for your professional comments. For this study, our hypothesis is that a protein located on the ribosome participates in the resistance of cotton to Verticillium wilt through phosphorylation. The goal of this study is to find the molecular mechanism by which GhRPS6 participates in the resistance to Verticillium wilt and to find the upstream and downstream proteins. However, it was found that the protein was localized in the nucleus during the experiment, and it was found that the protein had a toxic effect on yeast cells during the yeast two-hybrid screening interaction protein experiment. We have deleted the part of bioinformatics analysis, modified the remaining content, and added the discussion content of bioinformatics in the discussion part. In order to reduce the influence of exogenous genes on transgenic Arabidopsis thaliana, we left the terminator of the gene so that downstream GFP tags could not be expressed, so transgenic Arabidopsis thaliana could not be used for observation and localization. However, the experimental method of extracting Arabidopsis thaliana protoplasts is not fully understood and the transformation efficiency is not high. Tobacco cells are larger and considering the nuclear localization of proteins, we finally chose Nicotiana benthamiana for subcellular localization. We have checked the whole manuscript carefully for the grammar ortypo error, and reviewed the manuscripts in standard English. In addition, this manuscript has been edited by a native English professional with science background. We hope that it is acceptable for publish.

  1. Title: A ribosomal protein GhRPS6 was dentified??

Our response: Thanks for your careful reading. We have revised the Title.

  1. Line 129: showed that The, T in the should be small.

Our response: Thanks for your careful reading. We have revised the Text.

  1. Line 161: GhRPS6 silences reduced the … give a wrong meaning.

Our response: Thanks for your careful reading. We have revised this sentence.

  1. Line 169: proved to be no problem??

Our response: Thanks for your careful reading. We have revised this sentence.

  1. Line 174: incidence of silent plants was more higher, should be changed Our response: Thanks for your careful reading. We have revised the Text.

  2. Line 197: hr should be h
    Our response: Thanks for your careful reading. We have revised the Text.

  3. Line 202: And then, whose was increased??
    Our response: Thanks for your careful reading. We deleted this misdescription.

  1. Line 396 washed should be changed to rinsed.

Our response: Thanks for your careful reading. We changed the word.

  1. Line 399: there should be no space between the number and degree sign

Our response: Thanks for your careful reading. We have revised the Text.

  1. Line 404: the number should be written incorrect notation.

Our response: Thanks for your careful reading. We have revised the Text.

  1. Line 511: promoterin??

Our response: Thanks for your careful reading. The space bar is missing. We have already fixed it.

  1. Line 516: Arabidopsisplants??

Our response: Thanks for your careful reading. Here is a colloquial translation, we have realized the error and revised it into a technical term.

  1. Line 103: in the early proteomic analysis? Which analysis, this is the first result?

Our response: Thank you for your comments. Protein modification analysis is the result of previous laboratory experiments and has been published. The phosphorylation information of GhRPS6 comes from this experiment, so this result is mentioned here. Sorry for the wrong meaning, I will insert the reference in the part mentioned in the article.

  1. The genotyping results of both OX and silenced plants should be given, as a supplementary table.

Our response: Thanks for your professional advice. The genotyping results of silenced plants are shown in Figure 2b of the manuscript. We supplement the overexpression of Arabidopsis thaliana genotyping results presented in Supplementary Material 4B.

  1. How NO was measured was not clear. NO being highly reactive combines with other molecules to make SNOs or other RNS, it is therefore strange that how NO was exactly measured.

Our response: Thanks for your professional advice. Considering the high reactivity of NO, we only measured the dynamic content change within one hour by an Enzyme Linked Immunosorbent Assay Kit. And the Samples are taken in time for timely measurement. Three independent biological and technical repeats were performed for all samples. So that, the data trends are credible.

  1. Results section 2.1. how these analyses are related? Why the phylogenetic tree was constructed? Does the gene expression level of GhRPS6 has checked in cotton and its orthologous plants?

If the gene was cloned from the resistance cultivar? was the level of expression checked? The gene might already be overexpressed in the tolerant cultivar if it is involved in disease resistance, then why a 35S pro line was produced.

Our response: Thanks for your professional advice. First of all, protein modification omics analysis of plant roots of resistant and susceptible varieties of upland cotton after inoculation revealed that GhRPS6 phosphorylation (published results). Considering the complexity of upland cotton (Gossypium hirsutum) tetraploid and the high resistance characteristic of island cotton (Gossypium barbadense) to V. dahliae, these varieties were selected for evolutionary analysis. We also used transcriptome data to analyze the expression profile of GhRPS6 in upland cotton. The gene was cloned from the root cDNA of resistant cultivars, and its expression level was measured only when the silencing efficiency was tested. However, transcriptome data showed that the expression level of this gene was low in different tissues of upland cotton.

  1. Figure 2 A: the author should also show mock-treated plants for comparison.

Our response: Thanks for your professional advice. The Figure 2 A showed the Albino phenotype of TRV:GhPDS appeared 10 days after injection. This is as positive control to prove the reliability of VIGS.

  1. Figure 2, callose deposition and trypan blue staining needs to be repeated

Our response: Thanks for your professional advice. We modified the callose deposition and trypan blue staining. And three independent biological and technical repeats were performed for this.

  1. Figure 3 has no labels.

Our response: Thanks for your careful reading. We added a label to Figure 3.

  1. Figure 4 Pannel A and B are the both JA?

Our response: Thanks for your careful reading. We changed the annotation

  1. Figure 4: H2O2 induces NO production, then after 50 minutes post-inoculation, the TRV00 showed the highest H2O2 contents, while at the same time the same genotype showed reduced NO content. And if the NO contents data is correct, and If NOA is involved in NO synthesis, we would assume that high NO content in TRV00 after 60 minutes might be because of induced NOA expression to make more NO, however, the NOA expression in TRV00 at 1 hpi shows the opposite results. Can the author explains why?

The NO contents in TRV00 were high at 10 m reduced at 20, again raised at 30 and 40 and then reduced at 50. Can the author explain this zigzag nature of NO content in TRV00?

Does the c panel show DAB staining? The staining shows a clear difference in H2O2 level among the two plants after 24 h, however, it is not reflected in the data

Our response: Thanks for your careful reading. When we tested the outbreak of reactive oxygen species in GhRPS6-silenced and control plants within one hour after inoculation, the H2O2 content of silenced plants was lower than that of the control plants. An active oxygen burst is an important indicator of the initial stage of plant disease resistance. The in vitro experiments also confirmed that the control plants had more reactive oxygen bursts than the silenced plants. The content of NO in silenced cotton plants was slightly higher than that of control plants at the initial stage of infection, and then was lower than the control in the latter stage. The change in NO content may be due to the transient dynamic equilibrium between NO and H2O2.

  1. There is no coherence in the parameters. The NO, H2O2, contents were measured in a time interval of minutes with a maximum for 1 h, SA and JA contents were measured at 24 h, while the gene expression was studied up to 48 h. Hussain et al., 2016. doi: 10.3389/fpls.2016.00975. eCollection 2016 showed global changes in gene expression after 6 h of NO donor application. what was the logic to study NO and H2O2 contents that early?

Our response: Thanks for your careful reading. Reactive oxygen species change rapidly in plant stress response. And the H2O2 and NO is a dynamic equilibrium in plant cells. Many studies have shown that The NO, H2O2, contents were measured in a time interval of minutes with a maximum for 0.5 h. The expression of the gene is a dynamic process. So that, the gene expression is more commonly study up to 48-96h. While, the measurement of SA and JA were performed at 24 h according to the relevant literature.

Reviewer 3 Report

Verticillium wilt in cotton is complex. This manuscript describes one potential mechanism of resistance to that can be used in tandem with others strategies to develop much needed resistant cultivars.

'cultivar' and 'variety' are used interchangeably (including even in the abstract). Suggest changing 'variety' to 'cultivar' throughout the manuscript. Also, 'Verticillium' is capitalized - check throughout the manuscript including the abstract.

'uplands' first appears in Results on page 3, but is not identified as a colloquial term for species G. hirsutum until Materials and Methods page 14. Suggest introducing this in 2.1 Results (line 106), and indicating it is the most widely cultivated species globally. Also check throughout the manuscript that G. arboreum is not spelled arborum as it is in line 106 and Figure 1 title (it is correct in figure 1 legend).

Sentence line 71-73 indicates The ribosomal protein S6... is the first and only protein to undergo inducible phosphorylation in many years.' 'First' and 'only' are definitive terms, and 'in many years' indicates there were others before, please clarify.

Is there a reference for root-soaking method described in line 173?

Take care in Results 2.6 (lines299-302) to not overstate the conclusions - Verticillium wilt in cotton is complex and findings from this study are not likely to clarify the defense mechanism alone. It can be one of many strategies, genetic and agronomic.

Can a reference be included to verify the resistance of cv. Zhongzhimian2? Include a cultivar registration citation or similar identification. Also for Vd080.

Author Response

Reviewer #3: Verticillium wilt in cotton is complex. This manuscript describes one potential mechanism of resistance to that can be used in tandem with others strategies to develop much needed resistant cultivars.

concerning opinions:

  1. 'cultivar' and 'variety' are used interchangeably (including even in the abstract). Suggest changing 'variety' to 'cultivar' throughout the manuscript. Also, 'Verticillium' is capitalized - check throughout the manuscript including the abstract.

Our response: Thanks for your professional advice. We have revised the Text.

  1. 'uplands' first appears in Results on page 3, but is not identified as a colloquial term for species G. hirsutum until Materials and Methods page 14. Suggest introducing this in 2.1 Results (line 106), and indicating it is the most widely cultivated species globally. Also check throughout the manuscript that G. arboreum is not spelled arborum as it is in line 106 and Figure 1 title (it is correct in figure 1 legend).

Our response: Thanks for your careful reading. We have introduced uplands as a colloquial term for species G. hirsutum in 2.1 Results and indicated it is the most widely cultivated species globally. Check and modify the spelling of G. arboreum throughout the text.

  1. Sentence line 71-73 indicates The ribosomal protein S6... is the first and only protein to undergo inducible phosphorylation in many years.' 'First' and 'only' are definitive terms, and 'in many years' indicates there were others before, please clarify.

Our response: Thank you for your professional advice. This quote is a reference to an article published in 2015, which now includes a year limit.

  1. Is there a reference for root-soaking method described in line 173?

Our response: Thank you for pointing out that we have modified the root-soaking method to root dipping method

  1. Take care in Results 2.6 (lines299-302) to not overstate the conclusions - Verticillium wilt in cotton is complex and findings from this study are not likely to clarify the defense mechanism alone. It can be one of many strategies, genetic and agronomic.

Our response: Thank you for your professional advice. We have deleted the content without result demonstration

  1. Can a reference be included to verify the resistance of cv. Zhongzhimian2? Include a cultivar registration citation or similar identification. Also for Vd080.

Our response: Thanks for your suggestion. We supplement and cite relevant literatures in the materials and methods.

Round 2

Reviewer 1 Report

The authors have addressed the comments arisen in the previous review round. The manuscript can now be accepted.

Author Response

The authors have addressed the comments arisen in the previous review round. The manuscript can now be accepted.

Our response: Thank you for your comments and professional advice

Reviewer 2 Report

The authros have significantly improved the manuscript in terms of Language issues. however, the scientific soundness is still in question. I suggest that author should clearly draw a story line and stick their analyses towards their hypothesis. There are still some grammatical mistakes e.g. line 131, previously reported?

Author Response

1 The authros have significantly improved the manuscript in terms of Language issues. however, the scientific soundness is still in question. I suggest that author should clearly draw a story line and stick their analyses towards their hypothesis.

Our response: Thank you for your professional comments. We have interspersed the discussion with hypotheses and our results to complete the study.

2 There are still some grammatical mistakes e.g. line 131, previously reported?

Our response: Thanks for your careful reading. We have checked the whole manuscript carefully for the grammar ortypo error, and reviewed the manuscripts in standard English. We hope that it is acceptable for publish.

Round 3

Reviewer 2 Report

I have raised some points based on the results presented and the methods adopted in my first revision. Based on my comments, the author needs to repeat some experiments as some of the results are in contradiction with the previous reports that are published in reputable journals and are quite consistent. For example

  1. Figure 4: H2O2 induces NO production, then after 50 minutes post-inoculation, the TRV00 showed the highest H2O2 contents, while at the same time the same genotype showed reduced NO content. And if the NO contents data is correct, and If NOA is involved in NO synthesis, we would assume that high NO content in TRV00 after 60 minutes might be because of induced NOA expression to make more NO, however, the NOA expression in TRV00 at 1 hpi shows the opposite results. Can the author explain why?
  1. The NO contents in TRV00 were high at 10 m reduced at 20, again raised at 30 and 40 and then reduced at 50. Can the author explain this zigzag nature of NO content in TRV00?

The authors could not provide satisfactory answerers to my queries.

In addition, the DAB staining after 24 h does not reflect the data.

Author Response

Comments and Suggestions for Authors

I have raised some points based on the results presented and the methods adopted in my first revision. Based on my comments, the author needs to repeat some experiments as some of the results are in contradiction with the previous reports that are published in reputable journals and are quite consistent. For example

Figure 4: H2O2 induces NO production, then after 50 minutes post-inoculation, the TRV00 showed the highest H2O2 contents, while at the same time the same genotype showed reduced NO content. And if the NO contents data is correct, and If NOA is involved in NO synthesis, we would assume that high NO content in TRV00 after 60 minutes might be because of induced NOA expression to make more NO, however, the NOA expression in TRV00 at 1 hpi shows the opposite results. Can the author explain why?

The NO contents in TRV00 were high at 10 m reduced at 20, again raised at 30 and 40 and then reduced at 50. Can the author explain this zigzag nature of NO content in TRV00?

The authors could not provide satisfactory answerers to my queries.

In addition, the DAB staining after 24 h does not reflect the data.

Our response: This was not “H2O2 induces NO production“. In plant cell, it is a transient dynamic equilibrium between NO and H2O2 (https://doi.org/10.1104/pp.115.1.137, https://doi.org/10.3389/fpls.2016.00230 ). And, NO and H2O2 could lead to the cellular transition towards oxidative status.

An active oxygen burst is an important indicator of the initial stage of plant disease resistance. The pathogen can adhere to the root surface, and its hyphae penetrate roots to colonize the cortex. So that, we tested the outbreak of reactive oxygen species in the roots of GhRPS6-silenced and control plants within one hour after inoculation in the figure 4e,f. the results confirmed that the H2O2 content of silenced plants was lower than that of the control plants. And, “zigzag nature of NO content” may be due to the transient dynamic equilibrium between NO and H2O2 in plant cell.

Later, the hyphae then extend into the xylem and leaves. So that, Reactive oxygen species (ROS) was tested in leaves of silencing plants and control plants by DAB staining at 24 hours after inoculation. The results showed that the leaves of control plants had deeper staining and larger staining area, which demonstrated that the ROS levels were higher in leaves of control plants than that in the leaves of silenced plants. In addition, because of NOA is involved in NO synthesis, NOA expression in cotton leaves was performed by qPCR.

In a word, “after 50 minutes post-inoculation, the TRV00 showed the highest H2O2 contents, while at the same time the same genotype showed reduced NO content” due to “transient dynamic equilibrium between NO and H2O2”.

In addition, the pathogen can adhere to the root surface, and its hyphae penetrate roots to colonize the cortex. Later, the hyphae then extend into the xylem and leaves. In this study, the NO and H2O2 contents were tested from roots of cotton. While, the DAB staining and NOA expression was performed on leaves of cotton.
